# Do Live Weight, Body Condition Score, Back Muscle or Back-Fat Reserves Create the Suspicion of Goats Infected with *Eimeria* or Trichostrongylids?

**DOI:** 10.3390/ani11123591

**Published:** 2021-12-18

**Authors:** Martin Ptáček, Iveta Angela Kyriánová, Jana Nápravníková, Jaromír Ducháček, Tomáš Husák, Alfonso J. Chay-Canul, Claudia Zaragoza-Vera, Luis Cruz-Bacab, Jaroslav Vadlejch

**Affiliations:** 1Department of Animal Science, Faculty of Agrobiology, Food and Natural Resources, Czech University of Life Sciences Prague, 165 00 Praha-Suchdol, Czech Republic; duchacek@af.czu.cz; 2Department of Zoology and Fisheries, Faculty of Agrobiology, Food and Natural Resources, Czech University of Life Sciences Prague, 165 00 Praha-Suchdol, Czech Republic; kyrianovai@af.czu.cz (I.A.K.); napravnikovaj@af.czu.cz (J.N.); husakt@af.czu.cz (T.H.); vadlejch@af.czu.cz (J.V.); 3División Académica de Ciencias Agropecuarias, Universidad Juárez Autónoma de Tabasco, Villahermosa 86280, Tabasco, Mexico; aljuch@hotmail.com (A.J.C.-C.); claudia.zaragoza@ujat.mx (C.Z.-V.); eliezer.cruz@ujat.mx (L.C.-B.)

**Keywords:** depth of musculus longissimus thoracis et lumborum, parasitic control, strongylid nematodes

## Abstract

**Simple Summary:**

Response of live weight, body-condition score (BCS) depth of musculus longissimus thoracis et lumborum (MLTL) and back-fat thickness after infections with *Eimeria* sp. (EIM) and strongylid nematodes (STR) was investigated in a selected flock of dairy goats throughout the lactation period. Regression–correlation analysis demonstrated a significant interrelation of BCS to EIM exposure (BCS = 2.752 − 0.666 × 10^−3^ × EIM; *r* = −0.198). Goat nutritional status was not significantly correlated with STR infection. A linear tendency (*p* = 0.092), however, was detected for the response of MLTL to STR infection. An increase in infection intensity of 1000 eggs per gramme was accompanied by a 0.3 mm decrease in MLLT.

**Abstract:**

Thirty goats of the breeds Czech Brown Shorthaired and Czech White Shorthaired and their crosses were randomly selected from a flock at a farm in the Czech Republic. All animals were monitored for one year at monthly intervals for their nutritional status (live weight, LW; body-condition score, BCS; depth of musculus longissimus thoracis et lumborum, MLTL; back-fat thickness, BT) and infection intensity with *Eimeria* sp. (EIM) and strongylid nematodes (STR). Regression–correlation analysis showed a possible interrelation of BCS with EIM infection. Analysis of muscle and fat reserves indicated that BT was better than MLTL in identifying EIM infection. Goat nutritional status was not significantly correlated with STR infection. A linear tendency (*p* = 0.092), however, was detected for the response of MLTL to STR infection. Results of this study indicated theoretical use of BCS for *Eimeria* identification and suggested some perspective of BCS for targeting animals infected by strongylid nematode. Validity of our results, however, was limited by number of observed animals managed under specific breeding conditions.

## 1. Introduction

Gastrointestinal (GI) nematodes, particularly those from the superfamily Trichostrongyloidea, and coccidia of the genus *Eimeria* are important etiological agents of parasitic diseases in small ruminants [1,2,3]. The most pathogenic *Eimeria* species are *E. ninakohlyakimovae* and *E. arloingi* [4]. More than a dozen species of GI nematodes are responsible for parasitic gastroenteritis that occurs naturally in grazing animals [5,6]. Typical signs of these parasitic infections are deteriorated health and lower production parameters such as live weight, milk yield, feed conversion, and reproduction. The control of GI nematodes relies strongly on the use of anthelmintics, which are often administered to animals with no supportive diagnosis. These treatments may lead to the development of anthelmintic resistance [2,7], which poses a global threat to the health of small ruminants. Anticoccidial resistance is currently not as common as anthelmintic resistance, but the danger of this threat has already been registered by Odden et al. [8].

A promising approach of current science is aimed at selective treatment, where only animals with clinical symptoms of parasitic infection are treated [9,10]. The FAMACHAscale [11], DISCO (diarrhoea scores) [12], faecal egg count, or disruptions in the nutritional status of animals [9,13] are methods used to determine the need for anthelmintic treatment. Investigating new approaches for targeting animals suffering from coccidial infection is undoubtedly a promising challenge for future research as well.

The nutritional status of an animal can be assessed by live weight (LW) or the amounts of muscle and reserves of body fat [14]. The amounts of muscle and fat can be assessed subjectively using the body-condition score (BCS) [15] or by direct ultrasound measurements of body muscle and fat tissue [16,17,18]. The subjective BCS method, based on palpating the region of the last dorsal and first lumbar vertebrae and adjacent tissues using a 6-point scale from 0 to 5 [19], was originally described for sheep [20] and was then adapted for dairy goats by Harvieu et al. [21]. Information about the response of the nutritional status of goats to parasitic infection is very rare and generalised. Cornelius et al. [22] and Laurenson et al. [23] reported the effects of nematode infection on sheep LW and BCS. 

Extensive knowledge of the response of particular traits of nutritional status after exposure to parasitic infection should therefore serve as a tool of flock management for targeting animals infected with specific parasites.

The aim of this study was to identify interrelations among LW, BCS, back muscle, and back-fat reserves on one side and infection intensity with *Eimeria* and strongylid nematodes in original Czech breeds of dairy goats on the other side. These relations were subsequently suggested to define the importance of particular characteristics of goat nutritional status for targeting animals with parasitic load.

## 2. Materials and Methods

All procedures performed with the animals were in accordance with the Ethics Committee of Central Commission for Animal Welfare at the Ministry of Agriculture of the Czech Republic (Prague, Czech Republic) and were carried out in accordance with Directive 2010/63/EU for animal experiments. All investigations were carried out using data routinely collected during normal practice, in accordance with approved methodologies for recording performance in the Czech Republic and in accordance with official law in the Czech Republic (Animal Protection Against Cruelty Act; Act No. 246/1992 Sb.).

### 2.1. Flock Management

The study was performed on a private flock managed under an organic farming system in the Usti region of the Czech Republic in 2018/2019 (280–320 m a.s.l., mean annual temperature of 9 °C, and mean annual rainfall of 370 mm). The animals were housed on deep litter in barn stables and had year-round access to pasture in fenced areas. The dominant plant species in the pasture were *Lolium perenne*, *Festuca pratensis*, *Festuca rubra*, *Poa pratensis*, and *Trifolium repens*. Estimates of the feeding rations throughout the year are reported in Table 1. All animals had free access to drinking water (ad libitum). 

Kids were separated from their mothers in February and reared artificially. Goats were milked twice a day (06:30 and 16:30 side by side in a milking parlour with a capacity of 20 pairs) throughout the period of lactation. Milking frequency in the last two months of lactation was reduced to once a day because the ability to produce milk decreased. Goat drying (approximately 12 weeks before kidding) was based on milk production and udder status. Total milk production was 820 L with 2.94 of protein, 3.01 of fat, and 4.40 of lactose percentages in evaluated flock during the normalised lactation period (per 280 days).

### 2.2. Sampling Procedure and Parasitological Methods

Thirty mature goats of the breeds Czech Brown Shorthaired and Czech White Shorthaired and their crosses were randomly selected from the base flock of 117 goats. The only criteria for selection were different age, breed, kidded in similar time (±3 days), and similar milk production in previous parities of lactation. All the selected animals were individually monitored for one year (February 2018 to January 2019) at monthly intervals; however, they were bred, housed, and milked together with the rest of the flock. No anthelmintic or anticoccidial drugs were administered to the animals during this period and the preceding two months. All animals were assessed for their nutritional status, and faecal samples were collected. The animals were weighed (LW, kg), their body condition was assessed (BCS; points) [20,21], and the depth of the *musculus longissimus thoracis* and *lumborum* (MLTL, mm) and back-fat thickness (BT, mm) in the area of the last *thoracic vertebra* [16] were measured using an Aloka 500 ultrasound machine (Hitachi Aloka Medical, Ltd., Tokyo, Japan) equipped with a 5 MHz linear probe (UST-5011U). BCS estimation as all ultrasound measurements were performed by one experienced person. Faecal samples for parasitological examination were obtained directly from the rectum and stored in labelled plastic bags at 4 °C until examination. These samples were subsequently examined within 24 h after their collection to quantify the parasitic load of *Eimeria* (EIM) and strongylid nematodes (STR). The Concentration McMaster method [24] with a sensitivity of 20 parasitic elements (OPG-Oocysts Per Gram/EPG-Eggs Per Gram) was used for this identification. The prevalence of parasitic elements (eggs and oocysts) was evaluated as described by Bush et al. [25]. Prevalence was expressed as the number of goats infected with one or more individuals of either parasite (or taxonomic group) divided by the number of hosts. The intensity of infection was expressed as the number of eggs and oocysts in an infected host.

Parasitic elements obtained from the faeces were morphologically identified as described by Taylor et al. [26,27]. Precise morphological identification of most STR eggs to the genus level is virtually impossible, and therefore these eggs were only identified to the STR group. *Eimeria* oocysts were identified to species as described by Taylor at al. [26]. To obtain infective larvae for detailed strongylid identification, we collected pooled samples for larval culture and incubated them for seven days at 27 °C. The infective larvae were morphologically identified, as described by van Wyk and Mayhew [28].

### 2.3. Statistical Evaluation

All statistical procedures were performed using SAS software 9.4. General linear models were used for estimating linear regressions. Factors were selected on the basis of REGG procedure of STEPWISE method. On the basis of Akaike’s information criterion (AICC) and coefficients of determination (R^2^), finally, two statistical models were estimated for this evaluation:Y_ijklm_ = µ + SEASON_i_ + AGE_j_ + BREED_k_ + b × (EIM_l_) + e_ijklm_
Y_ijklm_ = µ + SEASON_i_ + AGE_j_ + BREED_k_ + b × (STR_l_) + e_ijklm_
where Y_ijklm_ is an evaluated trait (LW, BCS, MLLT, or BT), µ is the mean of the evaluated trait, SEASON_i_ is a fixed seasonal effect (i = February 2018, *n* = 30; i = March 2018, *n* = 30; i = April 2018, *n* = 30; i = May 2018, *n* = 30; i = June 2018, *n* = 30; i = July 2018, *n* = 30; i = August 2018, *n* = 30; i = September 2018, *n* = 30; i = October 2018, *n* = 30; i = November 2018, *n* = 30; i = December 2018, *n* = 30; i = January 2019, *n* = 30), AGE_j_ is a fixed effect of goat age (j = 3- and 4-year-old goats, *n* = 96; j = 5-year-old goats, *n* = 145; j = 6-year-old goats, *n* = 48; j = 7-year-old goats, *n* = 60), BREED_k_ is a fixed effect of the breed (k = Czech Brown Shorthaired goats, *n* = 96; k = Czech White Shorthaired, *n* = 24; k = Czech Brown Shorthaired and Czech White Shorthaired crosses, *n* = 228), b × (EIM_l_) is the linear regression for the traits of *Eimeria* sp. (range = 0–1420 OPG), b × (STR_l_) is the linear regression for the traits of the strongylid nematodes (range = 0–20,920 EPG), and e_ijklm_ is the residual error.

Pearson partial correlation coefficients were also used for expressing the relationships of the residuals for EIM and STR with the residuals for LW, BCS, MLLT, and BT. These correlations were estimated after adjusting the data for the effects of the annual seasonal effect, breed, and goat age

Significance levels of *p* < 0.05, *p* < 0.01, and *p* < 0.001 were used for evaluating the linear regressions and correlations.

## 3. Results

The variation of nutritional status of the dairy goats throughout the study period is shown in Figure 1.

The prevalence of parasitic infection (together with a description of the base statistical structure of the dataset) is presented in Table 2. The following *Eimeria* species were identified (prevalence, %): *E. arloingi* (58%), *E. aspheronica* (17%), *E. caprina* (8%), *E. ninakohlyakimovae* (7%), *E. christenseni* (5%), and *E. alijevi* (5%). The most prevalent strongylid nematodes were (prevalence, %): *Haemonchus contortus* (69%), *Trichostrongylus*/*Teladorsagia* (25%), *Oesophagostomum columbianum* (4%), and *Cooperia oncophora* (2%). The genera *Trichostrongylus* and *Teladorsagia* were merged due to their similar morphologies. All results indicated a minimal intensity of EIM infection that induced minimal health problems or loss of productivity. As we were aware of only minimal *Eimeria* infection in the evaluated flock, detection on statistical significance (Table 3) for this effect provided us the possibility to identify these relationships.

The results in Table 3 suggest the theoretical possibility of using BCS to identify animals infected with *Eimeria*. This suggestion was supported by the significance of the linear regression: an increase of 1000 OPG in the intensity of EIM infection corresponded with an approximate 0.7 decrease in BCS. This relationship was followed by a significantly negative correlation between OPG and BCS (*r* = −0.198, *p* < 0.001). Further analysis of BCS indicated that reserves of body tissues represented by back-fat thickness (BT = 3.151 − 0.772 × 10^−3^ × EIM, *p* < 0.01; *r* = −0.195, *p* < 0.01) responded more conclusively to EIM infection than did the development of body muscle (MLLT = 23.277 − 3.403 × 10^−3^ × EIM, *p* < 0.05; *r* = −0.195, *p* < 0.01).

The response of BCS to nematode infection was not significant but tended to decrease as infection intensity increased. An increase of 1000 EPG corresponded with a nonsignificant (*p* = 0.065) decrease in BCS of 0.035 (Table 4). Interestingly, separate investigations of MLLT and BT reserves indicated a more conclusive response of back-muscle reserves to STR infection than did fat reserves. The possible importance of MLLT was demonstrated by a tendency to decrease linearly; an increase of 1000 EPG corresponded with a decrease in MLLT of 0.3 mm (*p* = 0.092 in the model equation). Assessing goats for LW was not a good strategy because LW did not respond to either STR or EIM infection.

## 4. Discussion

Year–seasonal variation in nutritional status characteristics were previously demonstrated in dairy goats by Eknæs et al. [29] and Dønnem et al. [30]. This factor covers milk production and level of nutrition at a specified time period. Year-seasonal effect was a very important factor in our study as well, significantly influencing all the evaluated nutritional status characteristics in all model equations. Individual milk production within control days was eliminated in our study, as animals were selected with regard to their previous milk production history and date of kidding. Age of goat or parity of kidding, similarly to breed of animal, are important internal factors influencing goat nutrition status [31,32]. In line with previous studies, these factors were detected as significant in the majority. Considering the effect of systematic factors in the model equation, we could estimate interrelations among nutritional status and parasite infection intensity as much as possible. *Haemonchus contortus* and *Eimeria arloingi* are very pathogenic parasites in high-infection intensities. Our study showed that these two parasites were exactly predominant in the set of animals evaluated.

In general, not all animals in a flock shed the same number of GI nematode eggs, i.e., the faecal egg count (FEC). About 10% of the animals shed more eggs (have a higher FEC), and most of the animals have an intermediate or lower FEC [33]. This phenomenon of animals naturally prone to parasitic infection was also obvious in our study, indicated by the excessively high maximal values of the basic statistics. Selective treatment of susceptible animals thus helps to minimise anthelmintic resistance [34]. The natural susceptibility of hosts should be manifested in their levels of health, production, and nutrition, which should serve as promising metrics for the identification of these animals in a strategy of flock management. Field trials in Australia involving large flocks have demonstrated the utility of individual animal drenching according to BCS. This strategy can enable a large proportion of animals to remain untreated with no substantial loss of production [34,35]. Our results suggested theoretical potential use of goat BCS for *Eimeria* infection. These responses were obvious, even at very low intensities of *Eimeria* infection and a small set of animals. Some recent studies of *Eimeria* infection, however, warn against the serious threat of resistance to anticoccidials [36,37]. For that reason, verifying these relations should be in the scope of further research interest. Our results of muscle and fat tissue suggest the more precise targeting for body-fat reserves in this sense. The precise ultrasound measurement of muscle and fat is not likely to be applicable to flock management. BCS estimation, however, could provide sufficient information, because both the muscle layer and the back-fat layer can routinely be detected by a skilled person.

Cornelius et al. [22] investigated the responses of LW and BCS in Merino sheep to anthelmintic treatment. Ewes with poorer body condition prior to lambing were more likely to benefit from this treatment. The response of LW, however, was inconsistent in their study. In contrast, Besier [38] suggested that LW was a more appropriate tool of flock management than BCS for infections with *Haemonchus contortus*. The positive effect of nutritional status was further indicated on the basis of the results presented by Bessell et al. [39], wherein the application of anthelmintics improved LW gains and BCS in goats in Tanzania and India. The study by Bessell et al. [39] was performed without supporting information about parasitic infection or extensive knowledge of anthelmintic resistance. Their results, however, identified a very interesting connection between parasitic infection and goat nutritional status. We investigated the response of nutritional status permanently exposed to known *Eimeria* and nematodes measured regularly throughout the year (at one-month intervals).

In contrast to Cornelius et al. [22] and Bessell et al. [39], the response of BCS to nematode infection in our study was not significant but tended to decrease as infection intensity increased. This tendency also applied to the muscle layer as a part of body reserves for monitoring complexly assessed estimates of BCS.

The severity of the diseases caused by *Eimeria* and strongylid nematodes is also influenced by the presence and mutual interaction of different species of GI parasites in the host [4] and by environmental factors [40]. Results of present study were limited in number of observed animals managed under specific breeding conditions. The verification of our results under different breeding regimens and the assessment of their validity for other breeds of goats or sheep should thus be within the scope of subsequent research in this area.

## 5. Conclusions

Anthelmintic and anticoccidial resistance represent a current health challenge for goat breeding sector as drugs against these pathogens are often administered to animals with no supportive diagnosis. For that reason, we investigated the interrelation of nutritional status characteristics on one side to intensities of infection with *Eimeria* and strongylid nematodes on the other side. The results of this study indicated theoretical use of BCS for *Eimeria* identification and suggested some perspective of BCS for targeting animals infected by strongylid nematode. Validity of our results, however, is limited by number of observed animals managed under specific breeding conditions. For that reason, some perspective aspects should be verified some promising results for future.

## Figures and Tables

**Figure 1 animals-11-03591-f001:**
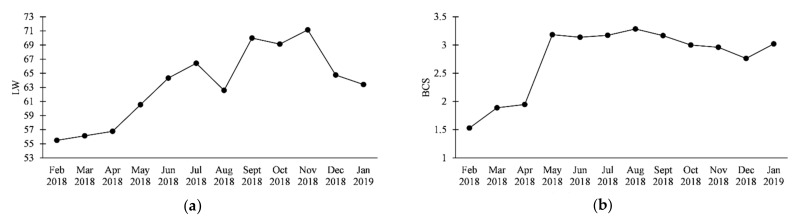
Seasonal variation of (**a**) mean goat live weight (LW, kg), (**b**) mean body condition score (BCS, points), (**c**) mean goat musculus longissimus thoracis et lumborum depth (MLTL, mm), and (**d**) mean back-fat thickness (BT, mm).

**Table 1 animals-11-03591-t001:** Components of the feeding rations estimated per goat per day.

	February April	May July	August October	November December
Grazing pasture	-	Ad libitum	Ad libitum	-
Concentrated supplement	2 kg	1.6 kg	1.4 kg	-
Haylage	6.1 kg	1.5 kg	1.5 kg	6.1 kg
Hay	ad libitum	ad libitum	ad libitum	ad libitum
Mineral licks	ad libitum	ad libitum	ad libitum	ad libitum

**Table 2 animals-11-03591-t002:** Characteristics of the infections with *Eimeria* and strongylid nematodes throughout the monitoring period.

Month		*Eimeria* sp.						Strongylid Nematodes				
	Prev ^1^ (%)	GM ^2^ (OPG ^7^)	AM ^3^ (OPG ^7^)	Sd ^4^ (OPG ^7^)	Min ^5^ (OPG ^7^)	Max ^6^ (OPG ^7^)	Prev ^1^ (%)	GM ^2^ (EPG ^8^)	AM ^3^ (EPG ^8^)	Sd ^4^ (EPG ^8^)	Min ^5^ (EPG ^8^)	Max ^6^ (EPG ^8^)
Feb	80	72	71	63.8	0	200	93	243	329	276.7	0	1140
Mar	65	36	28	28.9	0	100	100	455	836	998.2	20	4380
Apr	65	60	49	52.8	0	180	100	826	1298	1297.7	80	3040
May	43	34	19	29.0	0	120	100	3063	4541	4541.4	120	13,740
Jun	78	95	105	138.1	0	640	100	4308	6264	6264.4	340	20,980
Jul	32	115	64	63.9	0	180	89	1902	2536	2535.7	0	10,220
Aug	14	79	14	41.5	0	200	100	1146	1837	1837.1	40	4660
Sep	63	60	49	60.1	0	220	100	707	1200	1239.5	20	5500
Oct	64	115	148	302.4	0	1420	100	389	654	665.6	20	2840
Nov	92	184	260	275.6	0	1100	96	364	542	541.7	0	1880
Dec	83	90	103	122.8	0	540	83	122	144	144.4	0	540
Jan	70	62	52	52.5	0	180	83	190	226	226.1	0	740

^1^ Prev—prevalence; ^2^ GM—geometric mean; ^3^ AM—arithmetic mean; ^4^ Sd—standard deviation; ^5^ Min—minimal value; ^6^ Max—maximal value; ^7^ OPG—oocysts per gram; ^8^ EPG—eggs per gram.

**Table 3 animals-11-03591-t003:** Regression–correlation analysis between the characteristics of nutritional status with *Eimeria* corrected for defined factors.

Statistical Model of Partial Linear Regression	*p*-Value for Fixed Factors and for EIM as a Covariate in the Model				Pearson Partial Correlation (r)
	Season 1	Age 2	Breed 3	b*EIM 4	
LW ^5^ = 59.944 (±3.104) *** + 0.328 (± 3.703) × 10^−3^ × EIM ^n.s.^	<0.0001	0.0016	0.0007	0.9296	0.116 ^n.s.^
BCS ^6^ = 2.752 (±0.243) *** − 0.666 (± 0.308) × 10^−3^ × EIM *	<0.0001	<0.0001	0.1529	0.0314	−0.198 **
MLLT ^7^ = 23.277 (±1.299) *** − 3.403 (± 1.665) × 10^−3^ × EIM *	<0.0001	0.0019	0.0027	0.0422	−0.157 *
BT ^8^ = 3.151 (±0.209) *** − 0.772 (± 0.268) × 10^−3^ × EIM **	<0.0001	<0.0001	0.0015	0.0043	−0.195 **

^1^ SEASON—fixed seasonal effect; ^2^ AGE—fixed effect of goat age; ^3^ BREED—fixed effect of breed; ^4^ b*EIM—*Eimeria* sp. (oocysts per gram) as a covariate; ^5^ LW—live weight (kg); ^6^ BCS—body condition score points; ^7^ MLLT—depth of musculus longissimus thoracis et lumborum (mm); ^8^ BT—back-fat thickness (mm); *—significant at *p* < 0.05; **—significant at *p* < 0.01; ***—significant at *p* < 0.001; ^n.s.^—not significant.

**Table 4 animals-11-03591-t004:** Regression–correlation analysis between the characteristics of nutritional status with strongylid nematodes corrected for defined factors.

Statistical Model of Partial Linear Regression	*p*-Value for Fixed Factors and for STR as a Covariate in the Model				Pearson Partial Correlation (*r*)
	Season ^1^	Age ^2^	Breed ^3^	b*STR ^4^	
LW ^5^ = 60.209 (±3.061) *** − 0.359 (±0.265) × 10^−3^ × STR ^n.s.^	<0.0001	0.0013	0.0008	0.1763	0.032 ^n.s.^
BCS ^6^ = 2.681 (±0.241) *** − 0.035 (±0.019) × 10^−3^ × STR ^n.s.^	<0.0001	<0.0001	0.1931	0.0652	0.016 ^n.s.^
MLLT ^7^ = 23.092 (±1.295) *** − 0.338 (±0.200) × 10^−3^ × STR ^n.s.^	<0.0001	0.0007	0.0034	0.0922	−0.000 ^n.s.^
BT ^8^ = 3.090 (±0.211) *** − 0.037 (±0.033) × 10^−3^ × STR ^n.s.^	<0.0001	<0.0001	0.0032	0.2575	0.000 ^n.s.^

^1^ SEASON—fixed seasonal effect; ^2^ AGE—fixed effect of goat age; ^3^ BREED—fixed effect of breed; ^4^ b*STR—strongylid nematodes (eggs per gram) as a covariate; ^5^ LW—live weight (kg); ^6^ BCS—body condition score points; ^7^ MLLT—depth of musculus longissimus thoracis et lumborum (mm); ^8^ BT—back-fat thickness (mm); ***—significant at *p* < 0.001; ^n.s.^—not significant.

## Data Availability

The data presented in this study are available on request from the corresponding author. The data are not publicly available due to their containing information that could compromise the privacy of research participants.

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
