# Peer review of "Do Live Weight, Body Condition Score, Back Muscle or Back-Fat Reserves Create the Suspicion of Goats Infected with Eimeria or Trichostrongylids?"

_animals, 2021, doi:10.3390/ani11123591_

Round 1
Reviewer 1 Report
Dear authors,
The manuscript title is “Do live weight, body condition score, back muscle or back-fat reserves identify goats infected with Eimeria or strongylid nematodes?” and it aims to describe in detail the response of LW, BCS, back muscle, and back-fat reserves to intensities of infection with Eimeria and strongylid nematodes in original Czech breeds of dairy goats.
The topic falls within the aims and scope of the journal. Although the manuscript is not very original, it has scientific quality.
Nevertheless, the very small size of the sample (n=30) of one only flock are a constraint of this study, and care should be taken with the analysis of results.
The limitations of this study should be clearly stated.
Some particular suggestions/comments will be done here:
- Line 2/3 - Title – I would not say “identify” as there are several other factors that may lead to variations in live weight, body condition score or back muscle/fat reserves; on the edge, it may create the “suspicion of”, suggest or infer
- Line 17 and along the manuscript – shouldn’t it be MLTL? Or maybe you have to change the name of the muscle to “musculus longissimus lumborum et thoracis” in order to be MLLT
- Line 24 – a final point is missing after “infection”
- Line 40 – delete Eimeria as it is already in the title you do not need to repeat the same words as keywords; also “gastrointestinal tract” does not make sense
- Line 59 – add Cabaret et al., 2006 to number references
- Line 91 – you must detail what do you mean with susceptible animals, it is not clear
- Line 188 – Eimeria instead of coccidia (there are several other coccidia beside Eimeria), in this line and several others
- Line 192 - Table 2 – Eimeria instead of Eimeria; The strongylid should not be merged as the authors found different genera with different pathogenic potential
- Line 195 – 203 – I think you should re-write carefully these lines, because the burden of Eimeria (even looking at MAX OPG) is insignificant, even if the statistics says it is significant, keep in mind that this is ONE flock of only 30 goats with very low OPG
- Line 220 – gram instead of gramme
- Line 223 – replace eimeriosis instead of coccidiosis (also in other lines when you mean infections caused by the Eimeria )
- Line 231 – pathogenic instead of dangerous
- Line 236 – higher instead of high
- Line 242 – 244 – as sheep behave totally different from goats concerning parasites, I do not think you should keep this citation here
- Line 250 – 252 – Due to the reasons I mentioned earlier it seems to me that these sentences are clearly excessive; not only because the very low number of OPG but because the sample is tiny, so a lot of prudence is needed; maybe that is why “To the best of our knowledge, no previous studies have demonstrated the direct responses of these nutritional characteristics to parasitic infection in dairy goats.”
Author Response
The reply can you find in attached file. Thank you.

Reviewer 2 Report
General Comments:
This manuscript evaluated potential correlations between monthly fecal egg/oocyst counts of naturally infected dairy goats and their monthly live weights, body condition scores (BCS), back muscle or back-fat reserves. These were evaluated over a one-year period. Haemonchus was the predominant trichostrongyle in the flock and Eimeria arloingi was the predominant coccidian. The only statistically significant correlation identified in this study was between BCS and coccidian oocyst counts. There was also a linear tendency identified between the depth of musculus longissmus thoracis et lumborum (MLLT) and trichostrongyle egg counts, but this was not statistically significant. The methods and data analyses are appropriate for the correlation design of this study. The trichostrongyle parasite load appeared to be sufficient for showing effects if they were present, but the coccidian load seemed a bit low. The authors assume that the correlation between coccidian loads and BCS was caused by these parasites effecting goat nutrition among the more heavily infected goats, but it is equally possible that the lower level nutrition in some of the goats increased their susceptibility to the coccidians. It is impossible to separate these two possibilities given the experimental design, and It is well established that coccidian loads are increased by various stressors to immune function. Lower nutrition levels could easily be one of those stressors. Their assumption that the parasite is causing the observed body characteristics is also implied in their interpretation of the linear tendency identified between MLLT values and trichostrongyle egg output. These assumptions are the weakest part of this manuscript and should be corrected before publication. The fact that the authors cannot determine whether the increased coccidian loads cause the lower BCS or whether the poorer nutrition caused both the lower BCS and increased coccidian susceptibilities probably doesn’t eliminate the potential usefulness of using BCS as a warning sign for low levels of coccidiosis in these goats. However, because of the numerous potential causes of low BCS, it cannot be used a diagnosis of coccidial problems. In the discussion section, the authors should clarify how producers might use BCS in their surveillance of coccidial problems.
While determining if these nutritional-status-characteristic correlate with levels of parasitism in goats has value, the diagnostic potential of the “positive” findings are grossly overstated and need to be precisely described. At best, lower that expected BCS might indicate the need to evaluate oocyst numbers or monitor diarrheal scores. Given the prevalence of Haemonchus, it’s surprising that FAMACHA scores were not recorded in this study. Given the lack of any significant correlation with BCS or LW for the trichostrongyle intensities, it would be appropriate to recommend the use of FAMACHA scores under these circumstances. The authors make several overstatements relative to the practical value of this study. They need to modify these statement to insure that they precisely represent the contributions made by this study.
Title:
- The title should be modified to better reflect the findings of the study. Virtually all goats are infected with trichostrongyles and coccidia (when highly sensitive assays are used), and this study used egg/oocyst intensity not prevalence for their correlations. This title implies diagnosis of infected animals (prevalence) instead of intensity. The title also implies that this study was evaluating the diagnostic potential for these measurements of goat body characteristics. Given the numerous factors influencing these measurements, they can only suggest the need for further investigations with high levels of parasitism only one possibility. The term “strongylid” should be restricted to members of the family Strongylidae. The title should use the term “trichostrongylid” for the nematodes involved in this study.
Simple Summary
- The second-to-last sentence (line 23-24) needs to be modified to correctly represent the practical implications of this study. The only “nutritional status characteristic” correlating with parasite number is BCS, and it did not identify “suffering from parasite infection”.
- The only promising result was the correlation between oocyst numbers and BCS. The predictive power for MLLT was weak and this approach is not at all practical for producers even if it could predict trichostrongyle infections. The last sentence needs to be modified to provide a realistic future direction or drop this sentence.
Abstract:
- The term “infections” (in line 30) should be changed to better reflect the actual measurements made.
- Line 31: They measured correlations not response and it was to intensity levels not exposure.
- Lines 33-34: This sentence is an overstatement of what information BCS can provide. It can’t identify goats “suffering” from coccidiosis. The precision of this sentence needs to be improved.
- The last sentence needs to be modified or deleted as described for the “Simple Summary”
Introduction:
- Eliminate “Cabaret et al., 2006) on line 59.
- The sentence in lines 56 and 57 is not necessary.
- The sentence in lines 61 and 2 is not necessary.
- Move the sentence in lines 63 and 64 to the end of line 55.
- The sentence in lines 76 and 77 is very confusing.
- The sentences in lines 78-82 do not seem necessary.
- Lines 83-84: There are evidences that all ruminants mobilize nutrients to fight parasitic infections instead of using them for building muscle and fat.
- The second-to-last sentence (lines 88-90) is not correct. This study only measures correlations and it cannot evaluate causes. The word “response” implies response.
- The last sentence is good because it more precisely describes the value of identifying correlations.
Methods:
- Lines 93-99: Since this statement is also made in the Institutional Review Board Statement, is it necessary to also include it here?
- Were each of these assessments for nutritional status made by the same person for all of the animals?
Results:
- Larval identifications for the trichostrongyles can only be made at the genus level and so the species should not be included in the list (lines 184-185).
- Table 2: There is still some controversy among researchers whether geometric means better represents aggregation populations than does arithmetic means. I recommend providing both.
Discussion:
- Lines 223-229: These sentences should be combined into the similar sentences found at the beginning of the introduction. They don't need to be repeated..
- Lines 230-231: This sentence is too general. Haemonchus and arloingi were the two predominant parasites in each category. The authors should focus on these.
- Lines 231-232: This sentence seems like an overstatement. They are only serious in large numbers. In this study they are barely affective production parameters and not causing clinical symptoms.
- Lines 233-234: This sentence is unnecessary.
- Lines 239: Figures and tables are generally not referenced in the discussion. This observation should have been noted in the results section.
- Lines 250 and 252: The authors only measured correlations and did not determine cause. Therefore, the term “response” should not be used in these sentences.
- Line 272: The goats were continually exposed to the coccidians and trichostrongyles. The phrase “after exposure” suggests that they are not exposed at the beginning by later exposed.
- Line 275: The authors only measured correlations and did not determine cause. Therefore, the term “response” should not be used in this sentence.
- Houdijk’s recommendation to increase protein supplementation to limit trichostrongyle infections if very different that claiming that muscle mass might be useful for identifying trichostrongyle infections, and this study did not confirm the effectiveness of protein supplementation for controlling trichostrongyle infections.
- The linear relationship between trichostrongyle intensity and MLLT did not indicate “that natural resistance against EIM infection could also be acquired by the enrichment of feeding rations with energy fodders” (lines 281-284).
Conclusions:
- The final sentence in the conclusion is an overstatement of the value of this study. It would be best to eliminate it.

Author Response

(The authors gave the same response as above.)

Reviewer 3 Report
Do live weight, body condition score, back muscle or back-fat reserves
identify goats infected with Eimeria or strongylid nematodes?
The manuscript by Ptáček and collaborators presents an observational study
aimed at describing associations between some indicators of nutritional
state and parasite infestation in dairy goats.
The manuscript is well written and the issue addressed is highly relevant
for improving the health and welfare of dairy goats, as wells as
to optimize process in the flocks and the dairy industry.
The authors collected multiple samples over time, which is a
good practice to increase the power of the study. However the
use of animals that were submitted to the same environment without
having a group of individuals to contrast the results may limit the
conclusions of the study, i.e., the study may help to find associations,
but not causal inferences. This is an important point that the authors
could bring up at any point in the discussion.
The methodological aspects of the study seems to be fair, however
I suggest that the authors be clearer when describe important
procedures such as the model's structure, the categories used to
model the data and the methods used to adjust the degrees of freedom
and to check the goodness of fit of the statistical models.
I think that the manuscript would be greatly improved if the authors
could modify the presentation of the results (figures and tables) and
discuss some results that were not mentioned in that part of the text.
* Introduction
L 50:
The word "symptom" is reserved for the practice of medicine in humans,
which can directly express how they feel or experience some condition.
A more appropriate term to be used in animals is clinical "sign", given that
veterinary diagnosis are based on the interpretation of behavioral responses
of animals to illness.
L 61-63:
This sentence seems to be out of context and has not relevant information.
The sentence lacks a minimal information about how high, low or important
are the cost of treatment for the animal production industry.
Please read again and reword if necessary.
L 63-64:
This sentence also seems to be out of context in this paragraph.
Please read again and reword, move or delete if necessary.
L 84:
Can you be more specific about what "build natural resistance" means?
L 90-91:
Is this a second objective of your study, or a discussion resulting
from your first objective?
Did you collect additional data to achieve this objective?
* Materials and methods
L 100:
Can you provide data on dairy production and about the total
number of goats in this flock?
L 116:
Did you isolate the goats selected for this study from the
other individuals in the flock, or were they sharing the same
spaces?
I suggest you include a brief description of how the animals
selected for the study were housed and whether or not
they were in the same group or pen, or were in groups mixed with
other individuals.
Did you choose the goats to be included in the study based in
some criteria?
e.g., parity, milk yield, previous disease history...
L 128-130:
Could you please reword this sentence?
This is the final statement of a long sentence, but it is not well
adjusted to the context.
Perhaps you should try to make a better connection with the previous
sentence or split the sentences to improve the communication of your idea.
L 142-145:
I couldn't fully understand this sentence. Please check it and reword
if necessary.
Do the mean and standard deviation values represent the milk yield
in the sample or in the flock?
Did you collect the data on milk yield of the goats at the same time as
the other variables were measured?
L 155:
It's not clear how many models you built and whether you used
multiple response variables per model (multivariate), or only one
outcome per model for each trait.
Please check again this and improve the writing for clarity if necessary.
L 156-163:
Given that you have selected 30 animals for the study, the meaning of "n"
in this paragraph represents the number of pseudoreplicated observations
(i.e, ~ 30 individuals * 12 months).
How did you account for the effect of pseudoreplication in your models?
Did you verify the goodness of fit of your models? Please include that
information in the manuscript.
* Results
L 173-174:
Can you elaborate on this?
Please make clear what "variability" means in the context of this study.
L 178-179:
I suggest that you present any measure of dispersion with the average in the
graphics.
Do you have any special reason to present the information using two "y" axis
per graphic?
This strategy is discouraged for some authors because it may impair the
interpretation of the graphics and may also suggest that there are
associations between variables when they not necessarily exists.
In case you are interested in showing associations between the variables,
run some statistics, or even if you want to plot the data in the same graphic
use side-by-side, multi-panel plots or some of the different options
available to map multiple variables in one graphic.
L 204 and L 217 (Table 3, Table 4 and elsewhere):
I suggest that you include the coefficients (mean and standard error or CI)
for all the variables used in the model.
I'm not sure why you built two statistical models/test to analyze the data.
It appears to be redundant.
Why did you use the correlation analysis?
Is it necessary to perform that analysis in order to test your hypothesis?
L 217:
Should it be "Table 4"?
* Discussion
L 222:
Why you didn't discuss some results of your study?
i.e., the associations with season, age and breed.
I think the readers are also interested in the inferences
that can be made of the associations with that variables.
L 244-247:
This sentence suggest that you assessed the associations between lactation
stage and LW, BCS, MLLT and BT, but you only used those variables in the
models testing associations with parasite infestation.
Do you think the decrease in BCS is the result of parasite infestation, or
is it triggered by the increased metabolic requirement due to milk
production?
Please try to elaborate on this to make your argument clearer.
L 282-284:
Please reword for clarity.
* Conclusions
I suggest you improve the conclusions. Do you have an answer for
your hypothesis? Thus, put that information on the conclusions.
In addition (L 296) make it clear which criteria you suggest
should be used to identify susceptible animals.
Author Response

(The authors gave the same response as above.)

Round 2
Reviewer 1 Report
Well done, good job.
Congratulations!
Reviewer 3 Report
Dear authors,
Thank you for you response. I have no additional comments on your manuscript